# A Sign Language Recognition System Applied to Deaf-Mute Medical Consultation

**DOI:** 10.3390/s22239107

**Published:** 2022-11-24

**Authors:** Kun Xia, Weiwei Lu, Hongliang Fan, Qiang Zhao

**Affiliations:** Department of Electrical Engineering, University of Shanghai for Science and Technology, Shanghai 200093, China

**Keywords:** sign language recognition, deep learning, lightweight convolutional neural network, self-service for the deaf

## Abstract

It is an objective reality that deaf-mute people have difficulty seeking medical treatment. Due to the lack of sign language interpreters, most hospitals in China currently do not have the ability to interpret sign language. Normal medical treatment is a luxury for deaf people. In this paper, we propose a sign language recognition system: Heart-Speaker. Heart-Speaker is applied to a deaf-mute consultation scenario. The system provides a low-cost solution for the difficult problem of treating deaf-mute patients. The doctor only needs to point the Heart-Speaker at the deaf patient and the system automatically captures the sign language movements and translates the sign language semantics. When a doctor issues a diagnosis or asks a patient a question, the system displays the corresponding sign language video and subtitles to meet the needs of two-way communication between doctors and patients. The system uses the MobileNet-YOLOv3 model to recognize sign language. It meets the needs of running on embedded terminals and provides favorable recognition accuracy. We performed experiments to verify the accuracy of the measurements. The experimental results show that the accuracy rate of Heart-Speaker in recognizing sign language can reach 90.77%.

## 1. Introduction

Sign language is one of the main modes of communication between deaf and hearing people. When deaf people go to the hospital, they also need to communicate with doctors through sign language. However, most hospitals are not equipped with professional sign language interpreters, which makes it difficult for the deaf to seek medical treatment. There are two main problems [1]: First, it is difficult for deaf people to accurately express their physical symptoms to doctors, which can lead to doctors being unable to accurately grasp the specific etiology and symptoms of patients and it is easy to misdiagnose and miss a diagnosis. Second, patients can’t fully understand the doctor’s treatment suggestions and plans. Survey data show that [2,3] language consistency in the medical process can lead to better access to preventive services and precancer screening for deaf people.

In the past, due to technical limitations, people usually used remote connections to provide medical translation services for deaf patients [4]. However, the labor costs of such services are high and their reliability is affected by sign language interpreters’ acceptance of doctors’ technical solutions.

Sign language recognition refers to the use of algorithms and techniques used to identify the resulting sequence of gestures and articulate their meaning in text or speech. This technology involves many research fields, such as video acquisition and processing [5], computer vision, human–computer interaction, pattern recognition, natural language processing, etc. It is a challenging subject with high difficulty [6]. The development of this technology has reference significance for many research fields, such as video processing, computer vision, human–computer interaction, pattern recognition [7], and natural language processing. At present, the mainstream sign language recognition schemes can mainly be divided into two types [8]: sensor-based schemes and machine vision-based schemes. The authors of Ref. [9] proposed a wearable system for recognizing American Sign Language (ASL) in real time. By fusing information from an inertial sensor and sEMG sensors, this system’s sign language recognition accuracy is high. However, due to the high price of the sensors on the wearable system [10,11,12], the generalizability of this solution is limited.

With the rapid development of video capture, human–computer interaction and virtual reality technology, sign language recognition based on deep learning has received more and more international attention [13]. The addition of deep learning technology breaks through the bottleneck of sign language recognition through traditional methods [14]. Deep learning technology has better performance in both accuracy and robustness in sign language recognition systems. It creates a new method of human–computer interaction. The author of Ref. [15] discussed a system for tracking and recognizing gestures in real time based on depth data collected by Kinect. They proposed a new algorithm to improve scan time. The authors of Ref. [16] proposed a new character input system based on hand tapping gestures that can be used to facilitate human–computer interaction. This system used Kinect. Kinect is a line of motion-sensing input devices produced by Microsoft [17]. The devices generally contain RGB cameras and infrared projectors and detectors that map depth through either structured light or time-of-flight calculations, which can in turn be used to perform real-time gesture recognition and body skeletal detection, among other capabilities [18]. However, Kinect is expensive, so these methods are not suitable for low-cost products. In comparison, the method we used to collect sign language information through traditional cameras is more cost-effective.

The authors of Ref. [19] presented an approach that estimates 3D hand poses from regular RGB images. However, this system can only recognize gestures, not sign language. Sign language is a behavioral sequence formed by movements of the upper body and hands [20]. It has more flexibility and detail. The hand is the most flexible limb of the human body [21]. The distance and angle of the hand from the upper extremity and the movement of the five fingers affect the semantics of sign language [22]. At the same time, sign language behavior is easily affected by factors [23], such as background interference, illumination, observation angle, operator standardization, and so on. In existing datasets, operators usually only have upper limbs and hands moving. However, in practical applications, there are various situations, such as complex backgrounds, occlusion by multiple people, changes in lighting conditions, operator body movements, and unstandardized sign language, which makes identification more difficult.

Accurate and real-time sign language semantic recognition is the most important requirement for deaf people during consultation and a simple neural network model can no longer meet the system requirements. Although complex models can improve the network performance and the accuracy of sign language recognition, it can cause efficiency problems [24]. Due to the limited computing power and memory of the embedded terminal, the size of the model and the speed of detection can be limited. Therefore, it is necessary to select an appropriate neural network model to ensure the accuracy and speed of sign language recognition [25,26,27].

Of course, although deep learning technology has greatly improved the accuracy and speed of sign language recognition in recent years, it still cannot achieve the goal of complete real-time and accurate sign language recognition applications. Moreover, how to apply the existing sign language recognition technology to real life and realize cross-platform deployment is also an urgent problem to be solved [28].

The main objective of this research was to design a sign language recognition system based on deep learning to provide real-time and local translation for deaf people when they see a doctor. In order to make it acceptable to more people, the system should be low-cost and portable while maintaining the accuracy of sign language recognition.

To achieve these goals, we tried to use a lightweight sign language recognition model to process sign language videos captured by traditional cameras. This helps reduce costs and the amount of computation required for sign language recognition. Current sign language recognition solutions often require glove-style sensors or depth-sensing cameras, which make the entire system expensive. Most sign language recognition solutions require the use of a computer but we hope to integrate these functions into a small portable device so that deaf-mute people can easily take our device anywhere.

In summary, the contributions of this work are as follows:We designed a sign language recognition system specially used for deaf-mute medical consultation.The sign language is recognized by the MobileNet-YOLOv3 target detection model, which reduces the time required to recognize a single frame image while achieving high accuracy. Therefore, it can better realize real-time and efficient local target detection on embedded terminal devices.Heart-Speaker does not just recognize sign language. We also equipped it with the function of speech recognition so it can convert the voice information into corresponding sign language video and play it to deaf patients.We reduced the overall size and weight of the device so that people can take it anywhere. The system is equipped with a human body-sensing function. It automatically enters standby mode when there is no one in the detection area. This practice enhances the battery life of the device.We conducted a comprehensive analysis of application scenarios and proved that our system can effectively deal with various challenges in the medical treatment process of deaf people.Discussion on the various deep learning-based object detection techniques for sign language recognition is presented in Section 2. Section 3 provides our hardware design scheme and functional module design and introduces our program logic and system appearance. Section 4 introduces the sign language recognition model we used and the datasets applied for training and testing throughout experimentation. A detailed examination of the results and system performance is presented in Section 5. Section 6 discusses the significance of the proposed work. Section 7 provides the conclusions of the work with possible future orientations.

## 2. Related Work

### 2.1. Sign Language Recognition Based on Deep Learning

Due to the rapid development of deep learning, this technology has been increasingly used in gesture and sign language recognition research in recent years. Deep features tend to have a better spatial-temporal representation and can model more sign language actions and movement changes. Neural network models include CNN (convolutional neural network), RNN (recurrent neural network), GNN (graph neural network), etc. These have all been used in sign language research [29].

The main steps of sign language recognition are data acquisition, image preprocessing, detection and segmentation, feature extraction and classification [30]. After first segmenting and transforming the image, feature fusion of convolutional neural networks can be used to recognize sign language and extract features [31]. Commonly used image preprocessing methods include normalization, grayscale conversion, smooth filtering, noise reduction, and various morphological transformations. Median filtering and Gaussian filtering are common techniques for reducing image or video noise [32]. With the development of deep learning technology, some preprocessing methods have been incorporated into deep learning models. In sign language recognition research, the input image is usually resized before segmentation to reduce the computational load. Reducing the resolution of the input image can also improve computational efficiency [33]. Deep learning is trained through massive data [34], where the characteristics of the target sign language are automatically learned to complete the detection of the target sign language and the corresponding sign language segmentation is completed through the detected target sign language [35].

Sign language classification is classifying the extracted temporal and spatial features of sign language and it is the last stage in realizing sign language recognition [36]. The test dataset is classified and identified with the extracted sign language feature information. Common classification methods are DTW, SVM, HMM, and neural network-based methods [37]. In recent years, methods based on neural networks have made major breakthroughs in the fields of computer vision and image processing. Therefore, sign language recognition methods based on deep learning have become the mainstream [38].

In the application field of sign language recognition, the main problem [39] at present is that running deep learning models has high requirements on the processor performance of the platform, so it cannot perform well on some low-power embedded platforms. The authors of Ref. [40] presented a method for gesture recognition using a lightweight convolutional neural network. The authors of Ref. [41] used a neuromorphic camera (also called dynamic vision sensor, DVS) to capture sign language video and then used digital image processing algorithms to reduce redundant information.

### 2.2. Communication Problems between the Deaf and the Doctor

Few of the existing studies have targeted deaf people seeking medical treatment. At present, people still rely more on artificial sign language interpreters to solve the communication problems between hearing people and deaf people. However, in more private situations, such as in medical treatment, manual translation or third-party video communication are often inappropriate. This violates the patient’s privacy to some extent. The authors of Ref. [42] proposed a technology to map the remote sign language translation system so that the sign language translation support can be called at any time in the required scene. However, this method cannot solve the problem of two-way communication and it relies on the Internet, which would not provide real-time sign language interpretation support when the network fluctuates or there is no network service.

## 3. System Architecture

This section provides our system design choices.

### 3.1. System Overview

The architecture of the system framework is shown in Figure 1. The hardware of the sign language recognition system is mainly composed of MaixBit, a sign language information acquisition module, an intelligent voice module, a microwave radar module, a voice broadcast module, and an LCD display. As the core of the entire system control, MaixBit is used to schedule platform resources and run deep learning models. In this way, the system can realize the functions of single-target detection and non-contact wake-up of sign language. The intelligent voice module includes a voice recognition chip and a microphone. It is used to identify the doctor’s order; the wake-up of the system is realized by the detection of the microwave radar-sensing module; the power detection module monitors and manages the current lithium battery power. The battery can be charged through a USB Type-C interface.

### 3.2. Main Control

The main control has modules, such as a MaixBit (SIPEED, Beijing, China), ov7740 camera, voice recognition module, microwave radar sensor module, LCD module, and other modules. As the core of the entire system control, MaixBit realizes functions, such as single-target detection and non-contact wake-up in sign language. MaixBit is an open-source intelligent hardware development platform based on Canaan Technology Company’s Kendryte210 (K210, Canaan Inc., Beijing, China). Kendryte210’s central processing unit adopts a RISC-V architecture dual-core 64-bit CPU, equipped with neural network accelerator, and supports multi-modal recognition of machine vision and machine hearing. The power consumption of K210 is only 0.3 W, and the total computing power can reach up to 1TOPS. The chip comes with SRAM and an offline database. K210 can locally complete data processing and storage. It can be applied to various embedded development applications to meet actual needs.

The main control adopts the modular design idea to compile different functions into a clear data flow. The program can be divided into interface display, sign language information recognition subprogram, interactive subprogram of speech recognition module, intelligent broadcasting subprogram, and human body-sensing subprogram. Human sensing is set to the highest priority for fast sign language recognition when someone enters the detection range.

The flow chart of the main control is shown in Figure 2. When the microwave radar sensor detects someone directly ahead, it triggers the camera of the sign language recognition system and then MaixBit automatically recognizes the current sign language image by calling the trained deep learning model. At the same time, the system broadcasts to the doctor through the voice broadcast module and the system displays the current sign language meaning on the display screen in real time. Doctors can activate the voice recognition function of the system by means of keyword awakening. When the built-in microphone detects that the user has said, “start translation”, the system automatically exits the sign language recognition mode and enters the speech recognition mode. In this mode, the system recognizes what the doctor has said and searches the local video library for the corresponding semantic sign language video to play to the patient. When no one enters the recognition range, the system automatically enters the low-power mode to wait.

### 3.3. System Shell Structure Design

The mechanical structure design of Heart-Speaker follows a people-oriented principle. When the device needs to be carried around, the shell is required to be light and smooth and when the device needs to run for a long time, the shell is required to be safe and reliable with heat dissipation and insulation. Based on this, we designed the following mechanical structure design scheme. It consists of the bottom plate and the cover. The scheme is shown in Figure 3.

This work used Autodesk Inventor software for the overall modeling of the system, and used ABS-like stereoscopic light modeling resin materials. We used SLA 3D printing technology to process the outer shell. On the basis of ensuring the structural strength of the work, the weight is controlled to be below 300 g. The physical map is shown in Figure 4. The dimensional measurements of the device are 105 mm in length, 80 mm in width, and 20 mm in height. The thickness of the shell material is 2 mm. The chamfer radius on both sides of the device housing is 10 mm.

With reasonable planning of the internal space, the size of the system after assembly is comparable to a mobile phone. On the right side of the device are three buttons for power on, volume up and down, and mute. The back is provided with a hole for the camera and a hollow for the logo of the system. The hollow design is not only decorative but also used for external sensing of the radar module. At the same time, the edge of the shell is smoothed and polished, which provides users with a better holding experience. The screen is a 2.4-inch LCD display. Figure 5 shows the system size comparison diagram and the portability demonstration.

## 4. Sign Language Recognition Based on Machine Vision

The requirements of this system for the model are real-time and fast, so we need a low-latency and lightweight model. YOLOv3 is an end-to-end regression network model that can detect and classify at the same time. The MobileNet network is a deep neural network model with a separable convolution structure. The MobileNet-YOLOv3 network model was selected for sign language image recognition.

### 4.1. MobileNet-YOLOv3 Network Model

YOLOv3 (You Only Look Once version3) is an end-to-end target detection algorithm based on a convolutional neural network [43]. Its basic architecture, Darknet-53, is composed of a series of convolution layers with 1 × 1 and 3 × 3 convolution kernels. After extracting the feature information of the image through the Darknet-53 network, YOLOv3 uses k-means clustering to determine the bounding box. It predicts the bounding box through the feature layers of three scales after the basic network, thereby improving the detection effect of small objects. The last convolutional layer is the prediction layer. It outputs three tensor dimensions of bounding box, object location, and classification confidence. YOLOv3 incorporates the residual concept into the YOLOv2 network and draws on the shortcut layer of ResNet. Thus, the target detection effect close to the R-CNN series is achieved with fewer parameters and less computation.

MobileNet is a lightweight CNN model and the main component of the network structure is the depthwise separable convolution. It sets two hyperparameters, a width multiplier and a resolution multiplier, and designs a lightweight model that meets application requirements by adjusting the two hyperparameters [44]. A depthwise separable convolution decomposes a standard convolution into a depthwise convolution of $D_{K}\times D_{K}$ and a pointwise convolution of 1×1. Assuming that the input feature map has M channels and the output feature map has N channels, the size of the convolution kernel is $D_{K}\times D_{K}$ and the calculation amount of the depthwise convolution and the standard convolution is:$$\frac{H\times W\times M\times D_{K}\times D_{K}+H\times W\times M\times N}{H\times W\times M\times N\times D_{K}\times D_{K}}=\frac{1}{N}+\frac{1}{D_{K}^{2}}$$

As shown in Figure 6, it can be seen that the convolution method of the MobileNet network greatly reduces the amount of calculation compared with the standard convolution method. In the ImageNet dataset, the accuracy of MobileNet and VGG16 are almost the same but the number of parameters is only 1/32 and the amount of calculation is only 1/27.

The infrastructure of YOLOv3 is Darknet-53. It has a total of 73 layers of which 53 are convolutional layers and the rest are residual layers. The MobileNet network structure consists of 19 layers, including 17 layers of depthwise separable convolution and 2 layers of standard convolution. Based on the idea of the MobileNet model, the MobileNet-YOLOv3 model replaces the standard convolutional structure in the original YOLOv3 base network, Darknet-53, with a depthwise separable convolutional structure and then removes the fully connected layer and Softmax layer behind Darknet-53. The depthwise separable convolution module divides the convolution operation into two steps: depthwise convolution and point convolution. Among them, the depth convolution adopts different convolution kernels for different input channels for convolution and then completes the integration of the depth convolution output feature map through point convolution [45]. The reason for this is to avoid the defect that any convolution kernel in the ordinary convolution layer needs to operate on all channels. The network model established by the depthwise separable convolution structure has about 1/9 of the parameters of ordinary convolution. This greatly reduces the size of the entire model and greatly reduces the amount of computation.

### 4.2. Realization of Sign Language Recognition

#### 4.2.1. Sign Language Data Collection

To obtain adequate sign language recognition accuracy, a large dataset is required. However, due to the lack of sign language expertise and the high cost of annotation by most annotators, existing sign language sentence datasets usually only contain sentence-level labels and no lexical-level labels. Sign language emphasizes the temporal division and transmission relationship between gestures. The segmentation relationship refers to the reasonable pause points between independent words or phrases in a sentence; the transitive relationship refers to the reasonable order of appearance of the sub-visual elements that constitute gesture words under grammatical rules. Considering these factors, we chose the method of building our own dataset. We constructed a database as a sign language training set to facilitate research on sign language recognition using emerging deep learning techniques. We used an 8-megapixel camera as the image capture device for 19 sign language movements. These images were taken by five boys and five girls under different lighting conditions and different backgrounds. The five boys and five girls are made up of our four authors and six fellow students from our school’s sign language club. Sign language is a language that is not easy to learn, so we selected people with a basic background in sign language to participate in this research. Due to the particularity of the usage scenarios, data collection was carried out in a simulated hospital usage environment. Compared with the traditional datasets shot in the laboratory, the sign language datasets shot in simulated usage scenarios help to improve the detection accuracy of sign language recognition.

Data augmentation is common in data collection. It not only increases the noisy data but also improves the generalization ability and robustness of the sign language recognition model. The specific methods of data expansion are: rotation, vertical migration, horizontal migration, etc. We finally obtained a total of more than 4000 annotated sign language images, each containing more than 200 images. We divided these sample data into a training set, validation set, and test set according to the ratio of 3:1:1. Furthermore, to speed up image processing, these images were resized to 224 (pixels) × 224 (pixels) in the experiments. The sample data of the database are shown in Figure 7.

#### 4.2.2. Dataset Annotation

We chose to label the dataset by bounding boxes. Bounding boxes are one of the most common types of image annotations in computer vision. We use Labelelmg software for image annotation and an example of the annotation of the dataset is shown in Figure 8.

#### 4.2.3. Realization of Image Recognition Based on K210

We usually refer to the algorithms in the support model as operators. In theory, when these operators can be implemented in software, the model can be successfully run. If this model can be used in MaixPy programs, the program first needs to be able to run a .kmodel file and support the algorithms in the model. In this way, the input can be subjected to some calculation processes to obtain the output according to the description of the model.

The neural network model is computationally intensive and images with a large amount of data are used as input. Even if the CPU is used as the physical device for executing the software, the smooth calculation model cannot be satisfied. Therefore, we usually use a dedicated graphics accelerator card, namely the GPU, to accelerate graphics computing. The K210 is a very suitable GPU for sign language recognition. On the K210, a dedicated hardware KPU (Kendryte process unit) was designed, which is used for specific algorithms, so the operation speed is very fast.

K210 is equipped with a 400 MHz main frequency to support network operation. A lot of code for deriving models is integrated in MaixPy. Therefore, when using the KPU to calculate the target recognition model, only some internal functions need to be called without writing a lot of code.

Although the KPU has the ability to accelerate model calculation, it is limited by various factors, such as cost, power consumption, volume, heat generation, and application fields. It does not cover every type of operator; it only handles a part of them.

To infer a model on the KPU, the following requirements need to be met:
The number of channels is between 1 and 1024, the input feature map must be less than 320×240 (W×H), and the output feature map must exceed 4×4 (W×H);The model has the same symmetric padding (TensorFlow can use asymmetric padding when the stride is 2 and the size is even);Use any element-wise activation function (ReLU, ReLU6, Sigmoid, LeakyRelu, etc.); KPU does not support PReLU;The convolution kernel is 1×1 or 3×3 and the stride is 1 or 2.

Asymmetric paddings or valid padding convolution, transposeConv2D, Ordinary Conv2D, and DepthwiseConv2D (the convolution kernel is 1 × 1 or 3 × 3 but the stride is not 1 or 2) are available for KPU acceleration calculation.

A model is actually a set of structural and parametric data and different software can only recognize models in a specific format. KPU only recognizes models in the .kmodel format. The network trained by the computer is related to different frameworks. For example, TensorFlow is in the .h5 format or .tflite format. The network must be converted into .kmodel before it can be used by KPU and the nncase tool can achieve the purpose of model conversion. Due to the update of the nncase code, there are currently two main versions, V3 and V4. A V3 model is a model converted with nncase v0.1.0; a V4 model is a model converted with nncase v0.2.0. Comparing the two versions, V3 has high efficiency, less code, and less memory but supports fewer operators; V4 supports more operators but relies on software implementation, without hardware acceleration, and requires more memory. Both versions have their own advantages.

Figure 9 shows the loss and accuracy of the model as the number of iterations increases. As the number of iterations increases, the classification accuracy value eventually stabilizes at around 97%. After testing, the forward inference time per image is 70.1006 ms.

## 5. Experiments and Results

### 5.1. Sign Language Recognition Accuracy

#### 5.1.1. Test of Sign Language Recognition Accuracy

In the test session, a total of 19 key frames of sign language actions were collected, with a total of 4000 sets of sample data. Each sign language action included more than 200 test samples. The experimental environment is shown in Figure 10. The deep learning model used in the test experiment is implemented on the Ubuntu16.04 operating system based on TensorFlow and the GPU used is the NVDIA-3070. The training results are shown in Table 1.

#### 5.1.2. Accuracy Test under Different Training Networks

We trained and tested SSD, Faster RCNN, and MobileNet-YOLOv3 models on our homemade datasets. In the experiment, different parameters were used to debug the model. After analysis and comparison, according to the experimental results (training time, model fitting degree, model size, etc.), the optimal parameter settings are described next.

Since the size of the output frame of the network was fixed (a multiple of 32 × 32) and the original image was reduced to 1/13 of the original size after the sampling operation in the network, the size of the image at the input end of the network was set to 224 × 224 (for the training set, the image is cropped) and its input corresponding batch was set to 32. The number of iterations varies with the initial learning rate. The number of iterations is 60,000, 40,000 and 20,000 when the learning forces are $10^{-3}$, $10^{-4}$ and $10^{-5}$. In the process of optimizing the parameters, the optimization method used is the stochastic gradient descent method, the corresponding momentum is equal to 0.9, and the weight decay parameter is equal to 0.005.

Table 2 shows the comparison results of the MobileNet-YOLOv3 model used in this work and the current mainstream models, such as Faster RCNN, YOLOv3, SSD, etc., in the self-made test set. It can be seen from the table that, compared to the YOLOv3 method, the MobileNet-YOLOv3 model’s accuracy was improved by 5.2% and it was 20 FPS faster. This shows that the network model used in this work is improved compared to the original YOLOv3 model in terms of target detection accuracy and operation speed. It has better application prospects in real-time target detection and recognition scenarios.

### 5.2. Speech Recognition Test

The user can wake up each module of the system through the intelligent voice module. The user speaks an instruction to the intelligent voice module and the intelligent voice module captures the keywords in the instruction and sends the corresponding serial port data to the main control board, as shown in Figure 11.

We tested the recognition accuracy of the intelligent speech module in various environments with different noise backgrounds. The Heart-Speaker voice wake-up function was tested 300 times in each scenario. This experiment is used to verify that our system can have a favorable speech recognition success rate in the noise background of a hospital.

We integrated the experimental data and divided all the scenarios into three categories: below 30 dB, below 60 dB, and above 60 dB. The results show that the noisier the environment, the worse the recognition effect but the probability of completing the entire intelligent dialogue remains above 85%. The results are shown in Table 3. The experimental environment is shown in Figure 12.

## 6. Discussion and Significance of the Proposed Work

In view of the current situation that the deaf-mute group has difficulty seeing a doctor, we designed a sign language recognition system: Heart-Speaker for deaf-mute inquiries. Heart-Speaker adopts MaixBit as the core of the whole system control. The system can provide sign language interpretation for deaf patients in real time and can play the corresponding sign language video to the patients according to the doctor’s voice.

Heart-Speaker has a housing structure designed with portability and long-term use in mind. After reasonable planning of the internal space, the system includes the required components in the casing and, when fully assembled, it is the size of a mobile phone. The edge of the system shell is polished with rounded corners, which provides users with a better holding and carrying experience. Heart-Speaker has a 1500 mAh lithium battery, which can meet the long-term battery life requirements and can be charged through the USB Type-C interface. The system is also equipped with a microwave radar-sensing module, which automatically turns the screen off and enters standby mode when there is no patient in the detection area, further improving the battery life.

Due to the low computing power of the embedded image detection platform, the real-time demand for target detection in this project is relatively high. This system uses the MobileNet-YOLOv3 target detection model to identify the sign language image information collected by the camera, which enhances the model’s ability to detect small targets while maintaining high real-time performance. Therefore, it can better realize real-time and efficient target detection on embedded terminal equipment. MobileNet-YOLOv3 target detection model has strong adaptability and learning. It shows better performance in sign language recognition. We selected some sign language recognition systems in recent years to compare with our solution:

The authors of Ref. [46] used MediaPipe in conjunction with RNN models to address dynamic sign language recognition issues. It achieves a 99% recognition success rate for 10 words. Compared to this, our research results are nine percentage points lower in recognition accuracy but our system has faster recognition and can recognize more words. The authors of Ref. [47] proposed a deep-learning-based model that detects and recognizes words from a person’s gestures. The proposed model consisted of a single layer of LSTM followed by GRU. It achieves around 97% accuracy over 11 different signs. The authors made a 16-person Indian Sign Language dataset and posted it online. However, this system can only recognize gestures, not sign language. The authors of Ref. [48] proposed a dataset transformation system. The system consists of three steps: pose estimation, normalization, and spatial–temporal map (STmap) generation. Sign language recognition was implemented to evaluate the performance of this system. This approach greatly reduces the size of the dataset. Additionally, the sign language recognition accuracy is 84.5%. However, due to the use of pose estimation technology, it has high requirements on the computing power of the system and cannot be deployed on embedded platforms. The authors of Ref. [49] proposed an algorithm based on the palm definition model and linear models that can fully recognize 41 of the 42 letters in the Kazakh sign alphabet. The recognition accuracy of the system reached more than 90% and the cost is also quite low. The difference is that the former system focuses on identifying basic letters, while our system is more inclined to practical application scenarios. The authors of Ref. [50] proposed a receptive field-aware network with finger attention (RFaNet) that highlights the finger regions and builds inter-finger relations. This network achieves extremely high recognition accuracy with a American Sign Language spelling dataset. However, this system requires an input depth image, which is more expensive than our system.

## 7. Conclusions

On the one hand, this sign language recognition system provides deaf patients with a comfortable and simple medical experience and, on the other hand, responds to the call of the country for barrier-free construction. Computer vision technology is widely used and should be able to help deaf people better integrate into society and understand the world. After system testing, the system achieved positive results in various functions; has the advantages of high recognition accuracy, low-cost, convenient use, and intelligence; and has a wide range of application prospects.

In future work, we will expand to include more sign language vocabulary. In addition, we will try to develop a new function that can automatically save the collected sign language video during the sign language recognition process and use it as a sign language dataset to continue training our recognition model. At the same time, it can improve the present problem of an insufficient number of sign language datasets.

Improving the accuracy and efficiency of sign language recognition is also an important direction in our future research. We plan to iterate the object recognition model to the latest version of YOLO to further improve the accuracy of sign language recognition. In addition, we will also consider adding a depth camera to synchronously capture sign language video to assist in recognition. For hardware, we plan to use an independent chip to run the sign language recognition model so that the main chip of the system can obtain more computing power to expand the system functions, such as the automatic recording of sign language video mentioned above, connecting to the Internet to upload data, etc.

## Figures and Tables

**Figure 1 sensors-22-09107-f001:**
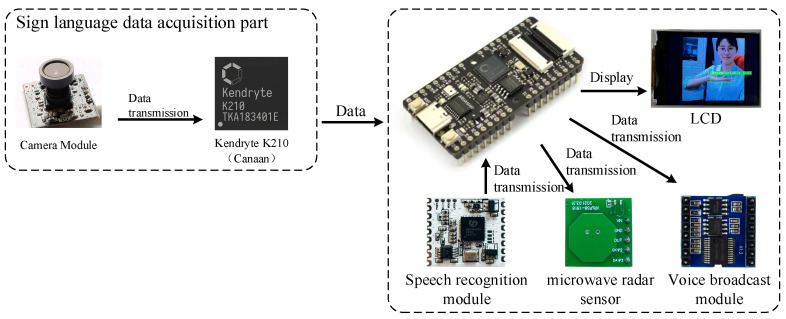
The architecture of the system.

**Figure 2 sensors-22-09107-f002:**
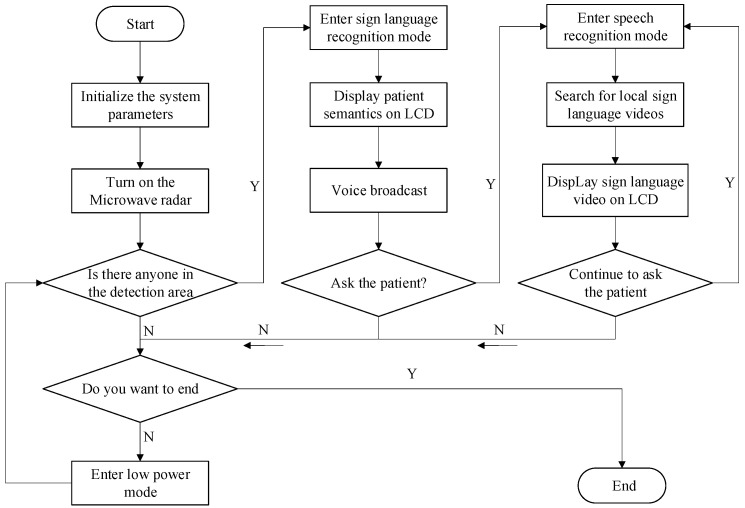
The flow chart of the main control.

**Figure 3 sensors-22-09107-f003:**
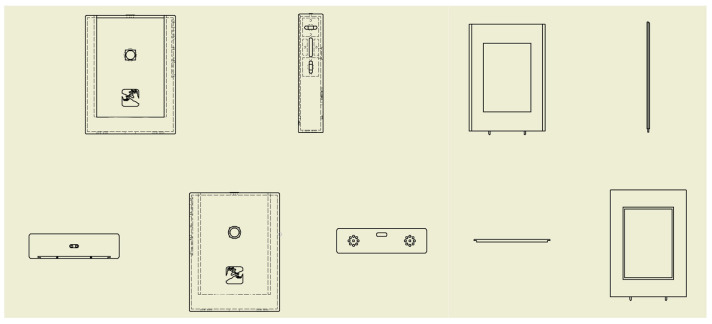
Heart-Speaker shell structure design.

**Figure 4 sensors-22-09107-f004:**
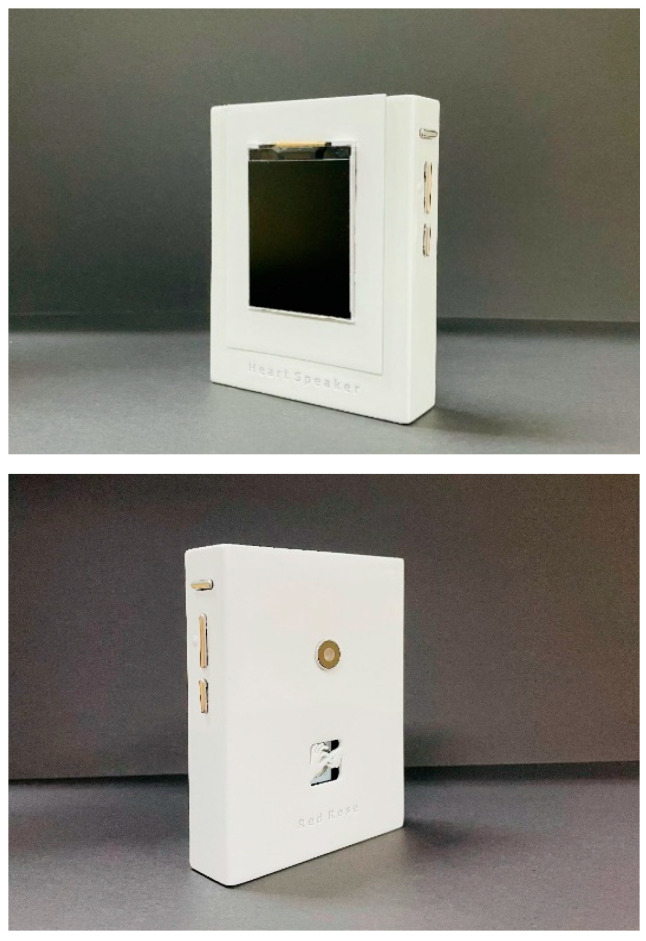
Physical map of Heart-Speaker.

**Figure 5 sensors-22-09107-f005:**
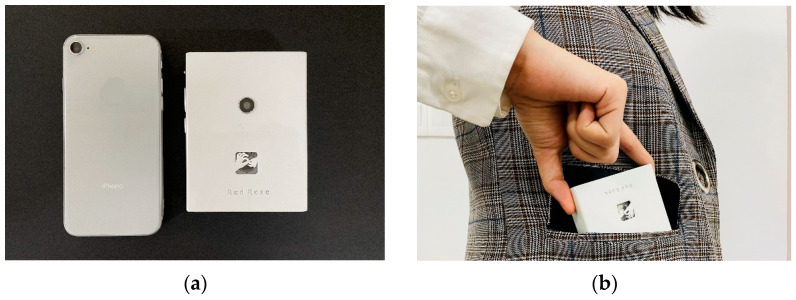
(**a**) Size comparison between Heart-Speaker and mobile phone; (**b**) Heart-Speaker’s portability demonstration as it can be carried in a jacket pocket.

**Figure 6 sensors-22-09107-f006:**
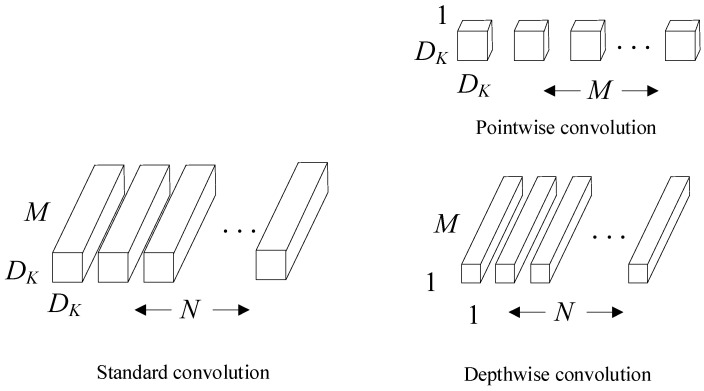
Standard convolutional structure and depthwise separable convolutional structure.

**Figure 7 sensors-22-09107-f007:**
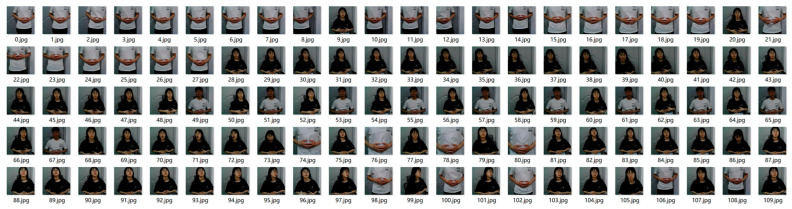
Annotation example of the database. These are the sign language that needs to be recognized.

**Figure 8 sensors-22-09107-f008:**
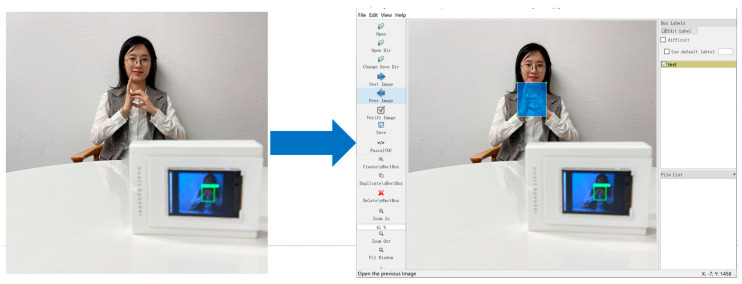
Sample data for the database.

**Figure 9 sensors-22-09107-f009:**
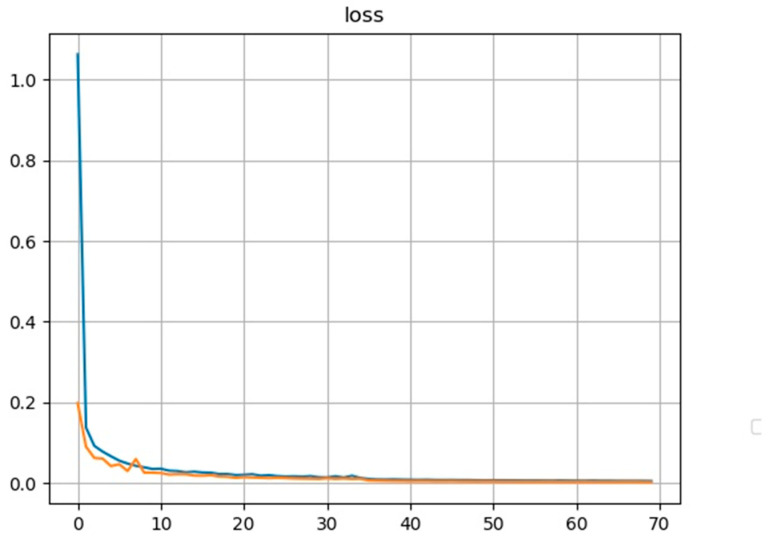
Model loss and accuracy curves as functions of iterations.

**Figure 10 sensors-22-09107-f010:**
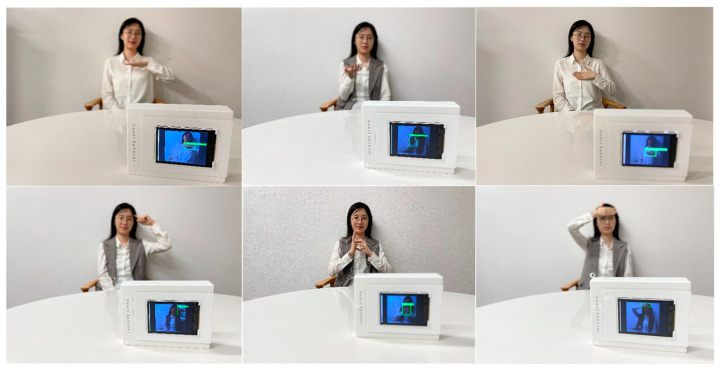
Scene graph of sign language recognition accuracy test. The environment was chosen in a room that mimics the decor of a hospital.

**Figure 11 sensors-22-09107-f011:**
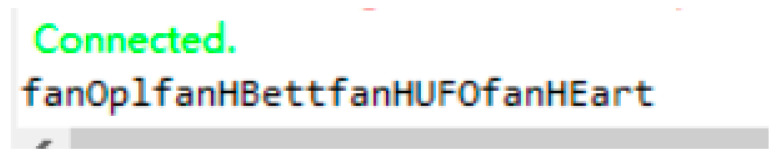
Serial port information successfully sent by the intelligent voice module.

**Figure 12 sensors-22-09107-f012:**
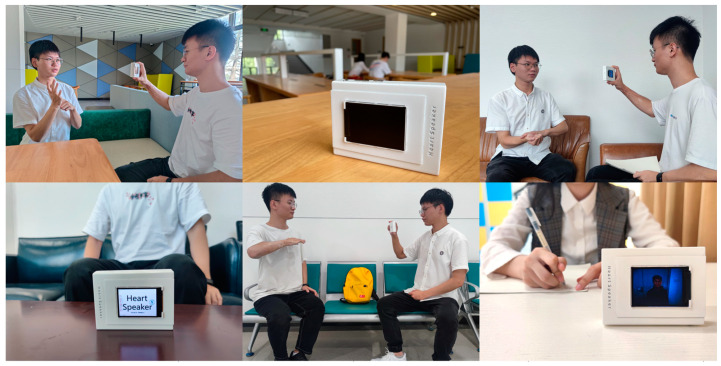
Different scenarios for testing speech recognition.

**Table 1 sensors-22-09107-t001:** The accuracy of sign language word recognition.

Word	Accuracy (%)	Word	Accuracy (%)
Belly	92.5879	Joint	89.3732
Cough	80.9091	Pain	89.8757
Fever (1)	92.3864	Sick leave (1)	96.2497
Fever (2)	89.3731	Sick leave (2)	80.9092
Fever (3)	91.4567	Runny nose	89.5687
Hand	90.5785	Shoulder	88.6153
Head	95.4689	Cancer	89.3698
Hello (1)	91.9972	Uncomfortable	90.9971
Hello (2)	90.6782	Want	90.5627
Me	92.1301		
Total Accuracy	90.77		

**Table 2 sensors-22-09107-t002:** Sign language recognition test results for different models.

Model	Basic Structure	FPS	mAP (%)
Faster RCNN	VGG16	7	65.2
SSD	VGG16	33	74.5
YOLOv3	Darknet-53	28	70.3
MobileNet-YOLOv3	Darknet-53	48	75.5

**Table 3 sensors-22-09107-t003:** Speech recognition test results.

Testing Scenarios	Testing Frequency	Errors Times	Mean Accuracy (%)
≤30 dB	300	2	99.33
≤60 dB	300	5	98.33
60 dB≤	300	27	91

## Data Availability

Not applicable.

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
