# Peer review of "A Sign Language Recognition System Applied to Deaf-Mute Medical Consultation"

_sensors, 2022, doi:10.3390/s22239107_

Round 1

Reviewer 1 Report

1.  What is the main objective of this research (please mention the main objective in the end of the background)

2. The research contribution is better to explain in paragraphs together with the state of the arts of each contribution. 

3. "These images were taken by 5 boys and 5 girls under different lighting conditions and different backgrounds" who is the participants of this research? it needs some detail explanation including the reason why they are chosen and why 5 boys and 5 girls are taken? why not more? is the gender matter? how about the severity level of deaf-mute they have? I think each deaf-mute person has  different level of severity. Has it been accommodated?

4. Please provide the discussion session by comparing this research results with other researches results

Reviewer 2 Report

This manuscript proposes a sign language recognition system applied to deaf-mute medical consultation. This system uses MobileNet-YOLOv3 target detection model to identify the sign language image information collected by the camera, which enhances the model's ability to detect small targets while maintaining high real-time performance  However, this manuscript has room for improvement, as follows:

 -This manuscript should discuss the problems to be solved and the disadvantages of the traditional schemes in current research.

 - Authors used MobileNet-YOLOv3 as target detection model. One such technique is to use YOLOv5 with Roboflow model, which generates a small size trained model and makes ML integration easier. Can you clarify how your model will be better?

 - Authors may include a new Section “Discussion and Significance of the Proposed Work” before Conclusion Section.

 -Incorporate more latest papers in your references.

Round 2

Reviewer 1 Report

Point 1: What is the main objective of this research (please mention the main objective in the end of the background)
-approved

Point 2: The research contribution is better to explain in paragraphs together with the state of the arts of each contribution

-approved and understood

Point 3: "These images were taken by 5 boys and 5 girls under different lighting conditions and different backgrounds" who is the participants of this research? it needs some detail explanation including the reason why they are chosen and why 5 boys and 5 girls are taken? why not more? is the gender matter? how about the severity level of deaf-mute they have? I think each deaf-mute person has different level of severity. Has it been accommodated?

-My question about the number and respondent background have been answered well. However, I am not sure about the involvement of authors as participants (as mentioned in the explanation that the four author are also the participants). Is it normal to involve author (researcher) in sign language research? How to make sure that the results wont be bias (free from subjective judgement of researchers/author regarding the expected result)?

-"As for why five boys and five girls were chosen because we wanted to balance the numbers of both genders to simulate real world situations" (are deaf-mute people in real world situation have the balance number of boys and girls?) how do you explain this?

Point 4: Please provide the discussion session by comparing this research results with other researches results.

-The discussion is not adequate as it only provides one  research that being compared with this research. At least 3 researches are cited in the discussion session and analyzed to be compared with this research results.
